# Shared genetic architecture between gastro-esophageal reflux disease, asthma, and allergic diseases
Tong Gong [1], Ralf Kuja-Halkola [1], Arvid Harder[1], Cecilia Lundholm[1], Awad I. Smew [1], Kelli Lehto[2], Anna Andreasson [3], Yi Lu [1], Nicholas J. Talley [4], Joëlle A. Pasman[1], Catarina Almqvist [1,5] & Bronwyn K. Brew [1,6] ✉

The aim is to investigate the evidence for shared genetic architecture between each of asthma, allergic rhinitis and eczema with gastro-esophageal reflux disease (GERD). Structural equation models (SEM) and polygenic risk score (PRS) analyses are applied to three Swedish twin cohorts ($n = 46,582$) and reveal a modest genetic correlation between GERD and asthma of 0.18 and bidirectional PRS and phenotypic associations ranging between OR 1.09-1.14 and no correlations for eczema and allergic rhinitis. Linkage disequilibrium score regression is applied to summary statistics of recently published GERD and asthma/allergic disease genome wide association studies and reveals a genetic correlation of 0.48 for asthma and GERD, and Genomic SEM supports a single latent factor. A gene-/gene-set analysis using MAGMA reveals six pleiotropic genes (two at 12q13.2) associated with asthma and GERD. This study provides evidence that there is a common genetic architecture unique to asthma and GERD that may explain comorbidity and requires further investigation.

Asthma is a common inflammatory respiratory disease causing acute dyspnea and wheezing, affecting 4-9% of the global child and adult population[1]. The most common non-allergic comorbidity of asthma is gastro-esophageal reflux disease (GERD), characterized by the reflux of gastric acid into the esophagus causing symptoms such as heartburn and regurgitation often leading to esophagitis and complications such as Barrett's esophagus[2]. GERD is predominantly an adult disease, although infants and adolescents can also suffer from GERD symptoms at clinically significant rates[3]. Epidemiological studies report the comorbidity of GERD in patients with asthma between 17% and 53% depending on the country of study (Western countries have a higher prevalence of GERD)[3], study population type (patient group or general population), and the detection methods used for asthma and GERD (symptoms alone or lab-based monitoring)[2,4,5]. Given that asthma affects 4-9% of the world's adult (18–45 years)[1] population and GERD 20–40% of the adult population[2], it is estimated that 1–4 in 100 adults around the world live with both asthma and GERD at any one time, varying by country and region.

GERD is well recognized to exacerbate asthma symptoms, particularly in those with severe asthma, and comorbidity has been shown to lead to worse quality of life, anxiety, and depression, compared to only having one of these illnesses[6–10]. However, despite this evident comorbidity burden, the origins of the co-occurrence of GERD and asthma remain unclear including whether a causal relationship exists[7]. Studies of adult and child populations for each disease have found evidence for a bidirectional association between the two diseases[4,11], and a recent Mendelian randomization (MR) study found evidence for a weak causal association between childhood (but not adult) asthma and GERD (Odds Ratio (OR) 1.003)[12]. On the other hand, a number of intervention studies aiming to improve asthma symptoms by using anti-reflux (acid-reducing) medication have not been successful[13,14].

There is some evidence to suggest that GERD is also associated with other atopic diseases including eczema and allergic rhinitis[15–17]. Eczema an inflammatory disease of the skin, and allergic rhinitis an inflammatory process of the nasal mucosa, as with asthma, are triggered by allergens and characterized by an IgE response[18]. Atopic diseases are characteristically childhood diseases beginning very early in life, with 20-25% of cases continuing into adulthood as well as new onset of adult cases particularly with asthma related to chronic smoking and chronic obstructive pulmonary disease (COPD)[18]. Asthma, GERD, allergic rhinitis, and eczema are complex

[1]Department of Medical Epidemiology and Biostatistics, Karolinska Institutet, Stockholm, Sweden. [2]Estonian Genome Centre, Institute of Genomics, University of Tartu, Tartu, Estonia. [3]Stress Research Institute, Department of Psychology, Stockholm University, Stockholm, Sweden. [4]School of Medicine and Public Health, University of Newcastle, Newcastle, NSW, Australia. [5]Pediatric Allergy and Pulmonology Unit at Astrid Lindgren Children's Hospital, Karolinska University Hospital, Stockholm, Sweden. [6]Centre for Big Data Research in Health & School of Clinical Medicine, UNSW, Sydney, NSW, Australia. ✉e-mail: Bronwyn.haasdyk.brew@ki.se

diseases with reported heritability of 53%, 26%, 55%, and 74%, respectively, based on a large meta-analysis of twin studies[19], and single nucleotide polymorphisms (SNP)-based heritability estimates explaining the population variation in GERD, asthma and allergic diseases have landed just above or below 10%[20,21].

We recently found evidence for a common origin of asthma and allergic diseases with GERD in a co-twin control study, supporting the hypothesis that a common genetic overlap may exist[15]. However, no studies have quantified the genetic overlap using diverse post-genome-wide association study (GWAS) analyses. Further investigation is needed harnessing recent summary statistics from GWAS and leveraging the rapid progress in polygenic risk score (PRS) and genomic structural equation modeling (genomic SEM) approaches to identify SNPs with the effects on cross-trait liability to GERD, asthma, and allergic diseases.

The aim of this study was to investigate the evidence for common genetic factors underlying the comorbidity between asthma, allergic rhinitis, and eczema with GERD using a triangulation of methods including both quantitative genetic approach (using classical twin modeling) and molecular genetic approaches (using PRS analysis, linkage disequilibrium score regression (LDSC), genomic SEM, and gene-based association tests).

Our study shows that by using quantitative twin modeling and molecular genetic techniques there is evidence for a modest shared genetic origin for asthma and GERD that may explain high rates of comorbidity. We also identify possible gene targets for further investigation. Little evidence is observed for GERD with eczema or allergic rhinitis.

## Results

Demographic and phenotypic information for each twin sub-cohort can be found in Supplementary Table 1. For the cohorts combined, the prevalence of asthma, allergic rhinitis, eczema, and GERD was 8.0%, 11.4%, 7.0%, and 12.0%, respectively. Bivariate phenotypic associations were confirmed for asthma and GERD-adjusted odds ratios (adjOR) 1.61, 95% CI 1.43, 1.80; allergic rhinitis and GERD-adjOR 1.14 (95% CI 1.02, 1.28); eczema and GERD-adjOR 1.14 (95% CI 1.00, 1.30) (Supplementary Table 2).

### Quantitative genetic analyses

Univariate results found higher correlations for GERD and all allergic diseases in monozygotic (MZ) twins compared to dizygotic (DZ) twins (Table 1), almost double in all cases suggesting substantial genetic components for each disease. We did not find any systematic pattern in the differences between males and females, except for opposite-sexed twins with the lowest intra-class correlations when compared with female or male DZ twins (Supplementary Table 3). Univariate quantitative genetic modeling found that AE was the best-fitting model with the lowest Akaike information criterion (AIC) for asthma and allergic rhinitis and ADE for eczema and GERD, suggesting a minimal role of shared environmental influences (Table 1). In the AE/ADE models, genetic influences explained 32% of the variance for GERD (95% CI 0.26, 0.39), 63% for asthma (95% CI 0.58, 0.67), 61% for allergic rhinitis (95% CI 0.57, 0.65) and 44% for eczema (95% CI 0.37, 0.51).

The phenotypic correlations (within individuals) between GERD and allergic diseases were moderate for GERD and asthma, $r = 0.14$; and weak for GERD and allergic rhinitis as well as GERD and eczema, $r = 0.04$ and 0.05, respectively. Cross-twin cross-trait correlations were small, but they were higher for MZ than DZ for GERD and asthma suggesting genetic influences, whereas correlations for GERD with allergic traits and eczema were close to zero (Table 2). Stratification by sex found similar results between males and females (Supplementary Table 3). The best-fitting bivariate model for GERD and asthma was the AE model (for parameter estimates for all models tested see Supplementary Tables 4, 5, and 6). About 56% of the covariance between GERD and asthma was explained by additive genetics and the other 44% was explained by unique environment (Supplementary Table 4). The decomposed genetic correlations were modest and positive between GERD and asthma $r_A = 0.18$, 95% CI 0.08, 0.28, and there was also a weak positive unique environmental correlation, $r_E = 0.11$, 95%

CI 0.03, 0.20 (Table 2). The phenotypic and cross-twin cross-trait correlations were too weak for GERD, allergic rhinitis, and eczema to estimate meaningful genetic and environmental correlations.

### Polygenic risk score analyses

Among the 26,895 twins with genotype data, we observed that PRS of asthma had a better prediction power than other traits (area under the receiver operating curve (AUC) range: 0.60–0.66, see Supplementary Table 7 and Supplementary Fig. 2). Associations between PRS and phenotypes for GERD and all allergic diseases are presented in Fig. 1 and Supplementary Table 8. The PRS of GERD was associated with asthma (aOR 1.14, 95% CI 1.08, 1.20 per one standard deviation (SD) increase in GERD-PRS), and similarly, we also observed the PRS of asthma being associated with GERD (aOR 1.09, 95% CI 1.05, 1.14 per one SD increase in asthma-PRS). There were no statistically significant associations observed for GERD with allergic rhinitis or eczema, except for GERD-PRS and allergic rhinitis which had an odds of 1.08 (95% CI 1.03, 1.13).

### Linkage disequilibrium score regression

SNP-based heritability results and LDSC genetic correlations are shown in Table 3 and Supplementary Table 9. Genetic correlations between asthma, eczema, and GERD were significant after Bonferroni correction ($r_g = 0.48$ asthma-GERD and $r_g = 0.20$ eczema-GERD, $p$-values < 0.01) but did not statistically significantly differ from zero for allergic rhinitis and GERD. Correlations were higher for adult asthma with GERD than childhood-onset asthma (COA) with GERD ($r_g = 0.33$, $r_g = 0.08$, respectively).

### Genomic SEM

A single latent factor fits the genetic covariance structure reasonably well, with comparative fit index (CFI) = 0.93 and standardized root mean square residual (SRMR) = 0.09. All traits loaded significantly on the common factor, with the strongest loading for eczema ($\beta = 0.85$, SE = 0.10, $p = 1E$-33) and the lowest loading for GERD ($\beta = 0.24$, SE = 0.03, $p = 2E$-14; Fig. 2, Supplementary Table 10). Using the combined asthma trait instead of childhood onset and adult onset separately deteriorated fit, with SRMR falling short of the <0.10 criterion (CFI = 0.96, SRMR = 0.15).

### Bidirectional two-sample MR analyses

There was support for a causal effect of genetic liability to asthma on the increased risk of GERD, which was consistent across different sensitivity analyses (Inverse variance weighted (IVW) OR 1.09, 95% CI 1.05, 1.14, $p = 6.55 \times 10^{-5}$). In the other direction, similar effect estimates of genetic liability to GERD on increased risk of asthma were also detected (OR 1.27, 95% CI 1.12, 1.43, $p = 2.65 \times 10^{-2}$). The MR-Egger regression intercepts did not significantly deviate from zero (Supplementary Table 11), suggesting no evidence of horizontal pleiotropy. Leave-one-out and Q-heterogeneity analysis showed that the effect estimates were not overly influenced by any one variant (Supplementary Figs. 3 and 4). No support for association between other allergic traits with GERD was observed (Supplementary Table 11, Supplementary Figs. 5–8).

### Gene-based association analysis

Using the Multi-marker Analysis of GenoMic Annotation (MAGMA) we identified genes significantly associated with asthma ($n = 352$), allergic rhinitis ($n = 2$), eczema ($n = 65$), and GERD ($n = 44$) (Supplementary Tables 12–17 (figshare/supplementarytables12-17) and Supplementary Figs. 9–12). After comparison, asthma and GERD were found to share six genes (*ERBB3, RBM6, HLA-B, SDK1, RERG, RAB5B*), one of which, *ERBB3*, was also significantly associated with both eczema and GERD. There were no other pleiotropic matches for eczema and GERD and none for allergic rhinitis and GERD (Table 4). Furthermore, we obtained regional association plots close to these six shared genetic loci from FUMA and confirmed that one common locus at 12q13.2 (where *RAB5B* and *ERBB3* are located) is associated with asthma and GERD (Supplementary Figs. 13–17). The independent SNP rs2069408 identified from the regional plot (risk allele G,

**Table 1 | Univariate twin correlations and twin model parameter estimates for GERD, asthma, and allergic diseases in all 3 twin sub-cohorts combined**

| | N (twin pairs) | | Twin correlations (95% CI) | | Model | Twin model estimates (95% CI)[a] | | | | |
|---|---|---|---|---|---|---|---|---|---|---|
| | MZ | DZ | rMZ | rDZ | | A | C or D | A + D | E | AIC |
| GERD | 5141 | 9374 | 0.33 (0.28, 0.37) | 0.11 (0.07, 0.15) | ACE | 0.30 (0.25, 0.35) | 0.00 (0.00, 0.00) | - | 0.71 (0.65, 0.76) | 21,388.85 |
| | | | | | ADE[b] | 0.12 (−0.10, 0.33) | 0.21 (−0.03, 0.45) | 0.32 (0.26, 0.39) | 0.68 (0.61, 0.74) | 21,385.89 |
| | | | | | AE | 0.30 (0.24, 0.35) | - | - | 0.70 (0.65, 0.76) | 21,386.85 |
| Asthma | | | 0.64 (0.61, 0.68) | 0.29 (0.25, 0.34) | ACE | 0.62 (0.53, 0.71) | 0.01 (−0.06, 0.08) | - | 0.37 (0.32, 0.42) | 15,693.01 |
| | | | | | ADE | 0.58 (0.34, 0.83) | 0.05 (−0.21, 0.31) | 0.63 (0.58, 0.68) | 0.37 (0.32, 0.42) | 15,692.73 |
| | | | | | AE[b] | 0.63 (0.58, 0.67) | - | - | 0.37 (0.33, 0.42) | 15,691.01 |
| Allergic rhinitis | | | 0.62 (0.59, 0.65) | 0.30 (0.26, 0.33) | ACE | 0.61 (0.57, 0.65) | 0.00 (0.00, 0.00) | - | 0.39 (0.35, 0.43) | 19,521.58 |
| | | | | | ADE | 0.60 (0.40, 0.81) | 0.01 (−0.21, 0.23) | 0.61 (0.56, 0.65) | 0.39 (0.35, 0.44) | 19,521.57 |
| | | | | | AE[b] | 0.61 (0.57, 0.65) | - | - | 0.39 (0.35, 0.43) | 19,519.58 |
| Eczema | | | 0.43 (0.38, 0.49) | 0.15 (0.10, 0.20) | ACE | 0.41 (0.34, 0.47) | 0.00 (0.00, 0.00) | - | 0.59 (0.53, 0.66) | 14,260.13 |
| | | | | | AD[b] | 0.18 (−0.10, 0.46) | 0.26 (−0.05, 0.56) | 0.44 (0.37, 0.51) | 0.56 (0.49, 0.63) | 14,257.42 |
| | | | | | AE | 0.41 (0.34, 0.47) | - | - | 0.59 (0.53, 0.66) | 14,258.13 |

*A* additive genetic component, *D* non-additive/dominant genetic component, *A + D* broad-sense heritability component, *C* shared environmental component, *E* non-shared environmental component (including measurement errors), *AIC* Akaike information criterion.
[a]Adjusted for sex and birth year (continuous, standardized).
[b]Best-fitted models with the lowest AIC.

**Table 2 | Bivariate association and quantitative genetic model estimates for GERD and allergic diseases in all three twin sub-cohorts combined**

| | GERD n (%) | Allergic diseases n (%) | Concordant pairs[a] n | Phenotypic correlations (95% CI)[b] | CTCT correlations (95% CI) | | Twin model estimates (95% CI)[b] | |
|---|---|---|---|---|---|---|---|---|
| | | | | | rMZ | rDZ | $r_A$ | $r_E$ |
| GERD & Asthma | 3420 (12.0) | 2257 (8.0) | 319 | 0.14 (0.11, 0.17) | 0.07 (0.02, 0.12) | 0.04 (0.00, 0.08) | 0.18 (0.08, 0.28) | 0.11 (0.03, 0.20) |
| GERD & Allergic Rhinitis | 3420 (12.0) | 3226 (11.4) | 403 | 0.04 (0.01, 0.07) | 0.02 (−0.03, 0.07) | 0.03 (−0.01, 0.07) | NA[c] | NA[c] |
| GERD & Eczema | 3420 (12.0) | 1984 (7.0) | 266 | 0.05 (0.02, 0.09) | 0.05 (−0.01, 0.10) | 0.02 (−0.02, 0.06) | NA[c] | NA[c] |

*CTCT* cross-twin cross trait, *MZ* monozygotic, *DZ* Dizygotic, *A* additive genetic component, *E* non-shared environmental component (including measurement errors).
[a]The number of pairs where one twin has GERD and the other twin has an allergic disease.
[b]Adjusted for sex and birth year (continuous, standardized).
[c]Phenotypic and CTCT correlation coefficients were too low to present twin model estimates, however, completeness estimates are presented in (Tables S4–S6).

**Fig. 1 | Associations between phenotypes and polygenic risk scores for GERD with eczema, allergic rhinitis, and asthma in Swedish twins.** The associations being presented in odds ratios (OR) are statistically significant when the horizontal line of the confidence interval (error bars) does not cross the vertical gray line at the value 1.

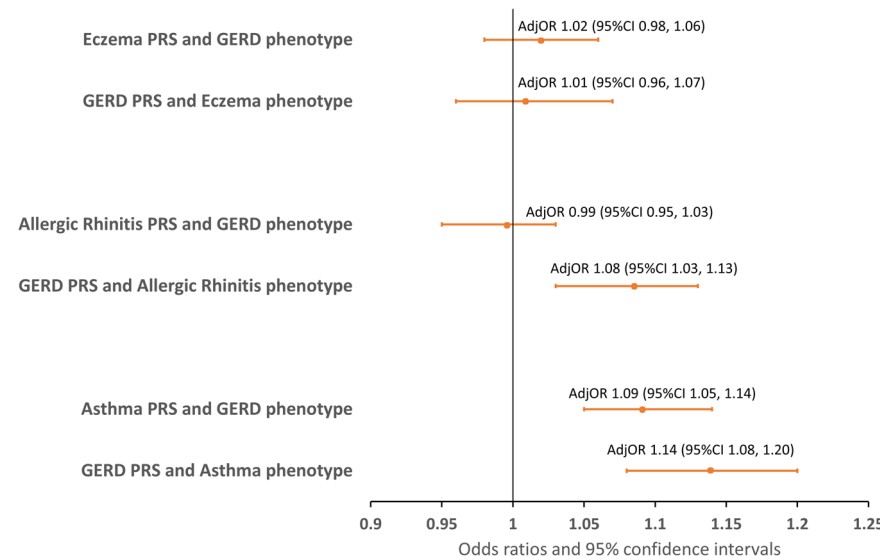

## Table 3 | Linkage-disequilibrium regression results between GERD and asthma, allergic rhinitis, and eczema

| Allergic diseases | N for allergic disease (sample prevalence %, population prevalence %)[a] | $h^2_{SNP\ (allergic\ disease)}$ (SE)[b] | Genetic correlation $r_g$ (CI) | p-values[c] |
|---|---|---|---|---|
| Asthma | 1,800,785 (8.5%, 8%) | 0.08 (0.004) | 0.48 (0.42, 0.53) | $7.83 \times 10^{-75}$ |
| Adulthood onset asthma | 327,253 (8.1%, 8%) | 0.13 (0.01) | 0.33 (0.27, 0.39) | $8.37 \times 10^{-26}$ |
| Childhood-onset asthma | 314,633 (4.4%, 5%) | 0.30 (0.03) | 0.08 (0.03, 0.12) | 0.0015 |
| Allergic rhinitis | 38,838 (27.2%, 25%) | 0.12 (0.02) | 0.15 (0.02, 0.27) | 0.0286 |
| Eczema | 796,661 (2.8%, 3%) | 0.08 (0.02) | 0.20 (0.11, 0.30) | $3.49 \times 10^{-5}$ |

[a]The population prevalence estimates were based on literature for most traits. However, we assumed a lower population prevalence of childhood-onset asthma to match the reported SNP-based heritability from the original GWAS.
[b]The SNP-based heritability for GERD, i.e., $h^2_{SNP\ (GERD)}$ and SE are 0.13 (0.01), based on the sample and population prevalence at 21.5% and 20% and the sample size of 332,601.
[c]Bonferroni corrected the significance level across 5 tested LD score regressions for the genetic correlation of allergic diseases and GERD at 0.01.

**Fig. 2 | Common factor (F1) model for all traits as estimated in Genomic SEM.** F1 is a latent common genetic factor of the genetic components of five GWAS phenotypes, i.e., adult-onset and childhood-onset asthma, allergic rhinitis, eczema, and GERD. The loading of adult-onset asthma was fixed to 1 for model identification purposes. One-headed arrows showed path regression estimates from the independent variable to the dependent variables. Standardized path estimates are given with their standard error in parentheses. The latent *u* variables with circular arrows reflect the residual variance in the genetic indicators not explained by the common factor.

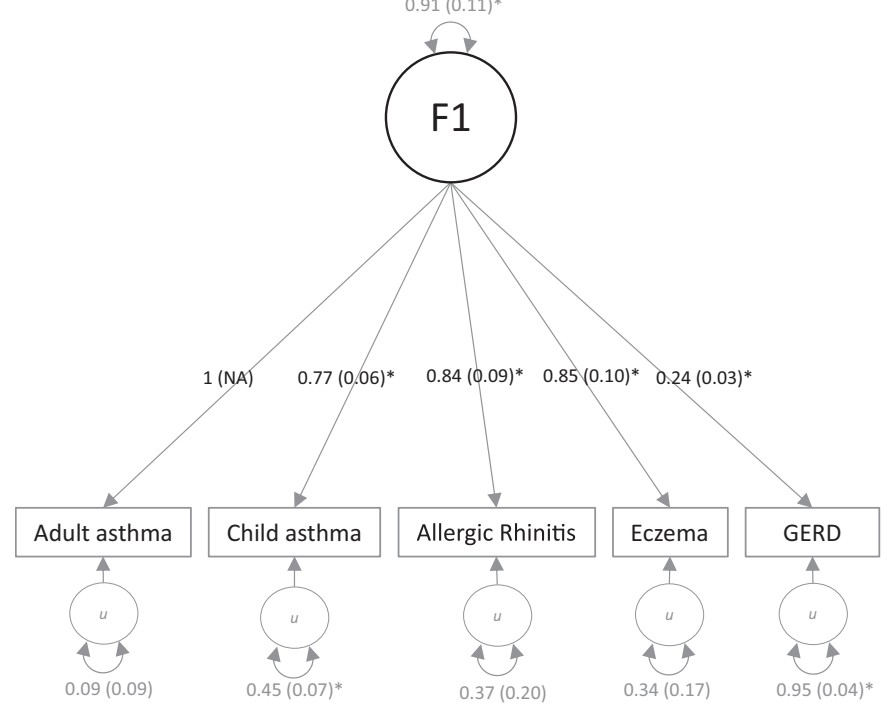

**Table 4 | The top significantly associated GERD genes identified by gene-based MAGMA analysis, which were also significantly associated with other allergic diseases after multiple-testing correction**

| Gene location | Gene symbol | Associated allergic diseases | p-value for GERD | p-value for the allergic diseases | Examples of associated traits from GWAS catalog searches |
|---|---|---|---|---|---|
| 12:56076799-56103505 (12q13.2) | ERBB3 | Asthma Eczema | 3.87 × 10⁻⁷ | 1.23 × 10⁻¹¹ (asthma) 9.70 × 10⁻⁸ (eczema) | Type 1 diabetes, BMI, education, math ability, moderate or severe asthma, smoking, |
| 3:4994007-50100045 (3p21.31) | RBM6 | Asthma | 1.41 × 10⁻⁷ | 3.05 × 10⁻⁷ | HDL, Body fat distribution, intelligence, BMI, education, insomnia, type 2 diabetes, income |
| 6:31353872-31367067 (6p21.33) | HLA-B | Asthma | 4.08 × 10⁻¹⁰ | 1.10 × 10⁻¹⁴ | Hip-waist ratio adjusted for BMI, blood protein levels, psoriasis, HDL, LDL, asthma, autism spectrum disorder or schizophrenia, eczema |
| 7:3301252-4269000 (7p22.2) | SDK1 | Asthma | 8.99 × 10⁻¹⁰ | 7.30 × 10⁻⁷ | Insomnia, math ability, smoking, lymphocyte, and eosinophil count, multisite chronic pain, Alzheimer's disease or GERD, risk-taking behavior, neuroticism |
| 12:15107783-15348675 (12p12.3) | RERG | Asthma | 6.33 × 10⁻⁸ | 4.85 × 10⁻⁷ | Glomerular filtration rate/creatinine levels, vaginal microbiome measurement, urate levels, education, red blood cell count, Alzheimer's disease or GERD, lung function (FEV1/FVC) |
| 12:55973913-55996683 (12q13.2) | RAB5B | Asthma | 2.10 × 10⁻⁷ | 1.61 × 10⁻⁹ | Asthma, hypothyroidism, HDL, Peptic ulcer or GERD drug use, FEV1, education, allergic disease (age of onset) |

The Bonferroni-corrected p-value thresholds for asthma, allergic rhinitis, eczema, and GERD are $2.52 \times 10^{-6}$, $2.57 \times 10^{-6}$, $2.53 \times 10^{-6}$, $2.62 \times 10^{-6}$, respectively.

asthma-GWAS $p = 5.397e{-}10$, beta = 0.0313) and the lead SNP rs11171710 (risk allele A, GERD-GWAS $p = 4.174e{-}9$, beta = 0.0351) were also positively associated with both asthma and GERD (Supplementary Fig. 13). Based on the GWAS catalog database, this locus is also associated with 269 traits including educational attainment, type 1 diabetes, body mass index, eosinophil and lymphocyte count, smoking, and hypothyroidism (Supplementary Table 16, figshare/supplementarytables12-17).

From all curated gene sets and GO terms obtained from MsigDB, we additionally identified hundreds of significantly enriched asthma-associated gene sets, 18 eczema-associated gene sets, and one GERD-associated gene set which is shared with the asthma-associated gene set: (GO_bp: go_positive_regulation_of_gene_expression, within the GO Biological Process aspect, $p_{GERD} = 0.002$ and $p_{asthma} = 0.0003$ after Bonferroni correction, Supplementary Table 17, figshare/supplementarytables12-17).

In the general tissue expression analysis, we found evidence for a small but significant enrichment exclusively in brain tissues for GERD-associated genes; blood, spleen, lung, and small-intestine tissues for asthma-associated genes; and spleen, blood, and small intestine tissues for eczema-associated genes, suggesting different tissues are responsible for GERD-gene and upregulated asthma-/eczema-gene signals, respectively (Supplementary Figs. 18–21).

## Discussion

This study explored the potential shared genetic origin between GERD with asthma, allergic rhinitis, and eczema using quantitative genetic and molecular genetic approaches applied to large twin datasets. We found consistent evidence of a modest genetic overlap between GERD and asthma across all methods, however, the analyses for allergic rhinitis and eczema with GERD revealed small to negligible genetic overlap.

We found that the phenotypic correlation for GERD and asthma was moderate at 0.14 and just over half of this was attributed to a shared genetic architecture. PRS and phenotypic associations were observed falling in the range of OR 1.09–1.14 per one SD increase, which indicates modest genetic overlap. The SNP-based genetic correlation was moderate at 0.48. The Genomic SEM results show that the common genetic liability to GERD, asthma, allergic rhinitis, and eczema can be summarized by a single underlying common factor. These findings confirm our earlier co-twin control and population-based analyses for GERD and asthma suggesting a common genetic origin[15], as well as the evidence for genetic correlation between asthma with GERD ($r_g = 0.40$) from a recent large-scale GWAS using global biobank meta-analysis initiative data[20]. In addition, there was a moderate unique environment correlation. Stronger correlations for adult rather than child onset asthma may help to point towards shared biological mechanisms of interest especially given that the genetic profiles for adult and child onset asthma have some differences[22]. However, it should be noted that a common etiology only accounts for a part of the comorbidity, which means direct pathways previously proposed may be important including microaspiration, neuroinflammation, or vagal reflex responses[23]. For example, clues may be found in recent studies using in vivo, ex vivo, and proteomic methods showing GERD to be associated with weakened bronchial epithelial barriers and increased inflammatory factors such as IL-33 upregulation[8,9].

Allergic rhinitis and GERD were weakly associated (phenotypic correlation = 0.04), and a genetic association between GERD-PRS and allergic rhinitis phenotype was supported by the PRS analysis. However, the lack of consistency in the direction of effects measured by the cross-twin cross-trait correlations, LDSC regression, and bidirectional MR analysis, suggests no evidence of a causal relationship between allergic rhinitis and GERD. Similarly, eczema and GERD estimates were null for bidirectional MR analyses, not supporting a causal connection either. Meanwhile, the LDSC regression revealed a possible signal for genetic overlap between eczema and GERD using summary data, which we could not replicate with individual-level data. Our previous study applying a co-twin control design also revealed only a weak association between self-reported GERD with allergic rhinitis and eczema[15]. Given that asthma, allergic rhinitis, and eczema have

shared genetic architecture[24], we would expect to find some shared genetic nature between the other two allergic diseases with GERD as we did for asthma-GERD but were not able to pick up the weak signals. This could possibly be due to the lack of power of the twin modeling and published allergic rhinitis/eczema GWAS. For example, the available summary data for allergic rhinitis from Waage et al was based on 38,838 individuals[25], including 10,563 cases, which was relatively smaller than the 153,763 asthma cases or 71,552 GERD cases (seen also in Supplementary Fig. 2 and Table 5). Further, differences in case recruitment (e.g., low prevalence in the UKB sample and high prevalence in the FinnGene sample) in the eczema GWAS meta-analyses may also have contributed to a diluted SNP-based heritability[26]. Therefore, although the triangulation of genetic methods in our study does not support a genetic explanation for GERD-eczema and GERD-allergic rhinitis comorbidity, future research harnessing larger, more accurately defined cohorts will provide further clarification.

The MAGMA analysis identified a distinct signal at locus 12q13.2 that is associated with both asthma and GERD and also corresponds to two pleiotropic genes (i.e., *RAB5B* and *ERBB3*) identified in the gene-based MAGMA results[21,27–29]. Locus 12q13.2 has also been reported to be associated with other obesity-related and autoimmune-related traits, although a precise biological pathway cannot be pinpointed by current gene- and gene-set-based results.

We confirmed the bidirectional causal association of asthma and GERD reported in previous studies[11,12]. The association between other allergic traits and GERD has seldom been studied, except for Ahn and colleagues who report a causal association between genetic liability to GERD with eczema using the same summary statistics[30]. Our null finding on GERD and eczema could be due to the stricter exclusion of any ambiguous SNPs being IVs (18 vs. 21 variants). The downstream evaluation by MR results (including one pleiotropic SNP at 12q13) did not show evidence for horizontal pleiotropy, suggesting that the putatively causal association between asthma and GERD was not biased by the two pleiotropic genes mentioned above, but it could still be mediated by them through possible inflammatory pathways (vertical pleiotropy). Further research is needed to tease out the specific role of 12q13.2 in GERD and asthma and the possible involvement of obesity, autoimmunity, and inflammation. In addition, our MAGMA analyses implicated another four pleiotropic genes—*RERG, RBM6, SDK1*, and *HLA-B* as potential targets for further investigation into asthma and GERD.

Using the MAGMA tissue enrichment analysis to detect differentially expressed gene sets for each disease we were expecting to find an overlapping tissue type for GERD and asthma or allergic rhinitis which may provide indication of a candidate gene set to explain genetic overlap. However, the results pointed to different tissue types, asthma gene sets were expressed in blood, spleen, small intestine, and as expected, in lung tissues. GERD genes were most expressed in brain tissues. Rather than shared expression in a shared tissue type, it may be that shared genes for asthma and GERD behave differently in each tissue for each disease, i.e., that gene in blood and spleen lead to inflammatory processes causing asthma, and that in the brain they play a sensory role in reflux pain. The brain is known to play an important role in behavioral traits such as smoking and food consumption choices which are important factors for GERD development[31]. Alternatively, expressed genes may be working through other pathways such as internalizing psychopathology disorders which are known comorbidities for both asthma[32,33] and GERD[34]. Indeed earlier work by this team and others looking at asthma and GERD comorbidity found that affective traits (depression, anxiety, and neuroticism) were important confounders of GERD and allergic disease associations[15].

The strengths of this study were firstly the size and scope of the study population and secondly, the triangulation of evidence from different approaches. We had access to three cohorts of the Swedish Twin Registry data covering a range of ages including children and adults over almost a whole century of birth dates. This improves the validity and generalizability of the study. Using a number of different methods harnessing both phenotypic and genotypic data from the twin cohorts allowed for triangulation

of evidence independent of each other. The consistent findings provide strong evidence for a genetic overlap at least in part for GERD and asthma. Finally, due to recent industrious efforts by other research groups to produce summary statistics from large GWAS studies and create gene repositories such as MAGMA, we were able to harness these to carry out polygenic risk score, LDSC regression, and gene-based association analysis to confirm and extend the quantitative gene analyses.

This study also has has some limitations. Firstly, the allergic rhinitis phenotype could be under-estimated in this study because the definition was based on a self-report of a doctor's diagnosis. Since allergic rhinitis often presents as a mild disease it may not be given a doctor's diagnosis and may be self-managed or with over-the-counter medications. Diagnosis of GERD was based on symptoms which are diagnostic but while the analyses would have been strengthened by also having objective test evidence of GERD (endoscopy and/or esophageal pH testing) this was not available. Secondly, the exploration of reported pleiotropic genes is based on very stringent p-value thresholds, which means we may have missed other potential target genes. Future studies should make use of other new and evolving analytical methods to investigate the strength of the pleiotropic genetic effects on both asthma and GERD at specific risk loci. Finally, there may be misclassification of phenotypes used in current GWAS due to the need to restrict time windows to boost sample size, thus diluting accuracy. As a consequence, in the context of heterogeneous and highly prevalent diseases like allergic rhinitis, eczema, and GERD, GWAS-identified common variants only explained a small part of the heritability compared to the moderate-to-high heritability found in twin studies.

In conclusion, using quantitative twin modeling and molecular genetic techniques we were able to show evidence for a modest shared genetic origin for asthma and GERD that may explain high rates of comorbidity. We also identified possible gene targets for further investigation. Little evidence was observed for GERD with eczema or allergic rhinitis.

## Methods
### Study participants
The study was based on phenotypic and genotyped data of participants from the Swedish Twin Registry (STR). The STR has collected extensive data on twins born 1886–2015 to study the genetic and environmental aspects of a broad spectrum of traits, phenotypes, and disorders[35,36].

In this study, all twins born in Sweden between 1911 and 1985 were invited to participate. Sub-cohorts included: the Study of Twin Adults Genes and Environment (STAGE)—born 1959-1985, response rate 59.6%, $n = 25,387$; TwinGene-born 1911–1958, response rate 46%, $n = 14,590$; and Screening Across the Lifespan of Twins—Young (SALTY)—born 1939–1958, response rate 65%, $n = 6605$. Information on zygosity was retrieved from either the collected DNA sample or the answers to five questions on twin similarity throughout questionnaires/interviews[35]. We excluded twin pairs if any twin's asthma, allergic rhinitis, eczema, or GERD information was missing or if only one twin participated in the study. In total, complete phenotypic data was available for 28,394 (61%) twins in 14,197 twin pairs.

DNA for polygenic risk score analysis was obtained at the study enrollment from saliva samples for STAGE and SALTY and from blood samples for TwinGene[37]. Individuals with DNA samples were genotyped using the Illumina Global Screening Array BeadChip, Illumina PsychArray BeadChip, and Illumina OmniExpress bead chip. Genotype imputation was performed using 1000 Genome data (Phase 3 Version 5) as the reference panel. Phasing was performed using Shapeit2 on each chromosome, and imputation was performed using Minimac3 on 5 Mb chromosomal chunks (with a window of 1 Mb on either side). After imputation, ~47 M markers were available, and over 7 M common variants (MAF ≥ 1%) have high imputation quality (imputation $R^2 \geq 0.8$).

During the quality control procedures of the genotype data, low-quality markers were removed, e.g., with call rates < 98%, that deviate grossly from the Hardy–Weinberg Equilibrium (p-value < 1e-6), large allele frequency differences from the 1000 Genome European reference samples, and

**Table 5 | Detailed information of published GWAS summary data of asthma, allergic diseases, and GERD used to calculate the polygenic risk scores and used in the LD score regression and genomic SEM analyses**

| Source | PMID | Phenotype | Sample size | Reference panel used | Statistical method used | SNP-based heritability reported | Summary data download URL |
|---|---|---|---|---|---|---|---|
| **Asthma** | | | | | | | |
| Ferreira MA et al.[22] | 30929738 | Childhood-onset (COA) and adulthood-onset asthma (AOA) | 447,628 (however, the UKB and QSKIN data is based on 327253 and 314633 individuals) | 1000 Genome Project | Linear mixed model + Logistic regression + inverse-variance-weighted fixed-effects meta-analysis | $h^2_{SNP}$ (COA) = 25.6%, $h^2_{SNP}$ (AOA) = 10.6% | https://genepi.qimr.edu.au/staff/manuelF/gwas_results/CHILD_ONSET_ASTHMA.20180501.allchr.assoc.GC.gz https://genepi.qimr.edu.au/staff/manuelF/gwas_results/ADULT1_ADULT2_ONSET_ASTHMA.20180716.allchr.assoc.GC.gz |
| Zhou W et al.[40] | Cell Genomics (in press) | Asthma | 153,763 cases/1,647,022 controls | 1000 Genome Project + Human Genome Diversity Project | SAIGE or REGENIE by cohort and inverse-variance weighted fixed effect model for meta-analyses | $h^2_{SNP}$ = 8.7% | https://github.com/globalbiobank |
| **Allergic rhinitis** | | | | | | | |
| Waage J et al.[25] | 30013184 | Allergic rhinitis (symptoms/ diagnosis) ever | 59,762 cases/152,358 controls. However, the downloaded summary data included 38,838 individuals | UK10K | Inverse-variance weighted fixed effect model for meta-analyses | $h^2_{SNP}$ = 7.8% | https://hmgubox.helmholtz-muenchen.de/d/b55da086360c40118ae8/files/?p=/2018-05-11_EAGLE_AR.txt.gz |
| **Eczema** | | | | | | | |
| Sliz E et al.[26] | 34454985 | Diagnosis of atopic dermatitis | 22,474 cases/774,187 controls | Finnish population-specific SISu v3 + Estonian-specific reference panel + 1000 Genomes phase 3 | Inverse-variance weighted fixed effect model for meta-analyses | $h^2_{SNP}$ = 5.4% | http://ftp.ebi.ac.uk/pub/databases/gwas/summary_statistics/GCST90027001-GCST90028000/GCST90027161/harmonized/34454985-GCST90027161-EFO_0000274.h.tsv.gz |
| **GERD** | | | | | | | |
| An J et al.[21] | 31346403 | GERD: Self-reported or diagnosis or medication | 71,522 cases /261,079 controls (excl 23andMe) | 1000 Genomes Phase 3 | Fixed effect model for meta-analyses | $h^2_{SNP}$ = 11.3% | https://figshare.com/articles/dataset/GERD_GWAS_summary/8986589 |

low-quality score, i.e., mean GenCall scores < 0.5. In total, we removed ~2% of samples with a sample calling rate < 98%; unusual heterozygosity; possible sample contamination; sex violation; or non-European ancestral outliers. After removing low-quality samples and imputing the genotypes of MZ twins from their paired genotyped twins, there were 26895 unique samples (9589 twins from STAGE, 6 398 twins from SALTY, and 10,908 twins from TwinGene).

All participants provided informed consent before participation, and the data was pseudonymized for management and analyses. Ethical approval was provided by the Swedish Ethical Review Authority. All ethical regulations relevant to human research participants were followed.

## GERD
Data were collected from questionnaire/interview data which included GERD-specific symptom questions[38,39]. GERD was defined as reporting 1) heartburn more than once per week ("heartburn symptoms"), OR 2) pain behind the sternum more than once per week, and the pain was relieved by antacid or acid-suppressing medicine ("reflux-like chest pain").

## Asthma, allergic rhinitis, and eczema
Data were collected from questionnaire/interview data which included questions about asthma, allergic rhinitis, and eczema. A case was defined as reporting a current disease and having received a doctor's diagnosis[24].

## Polygenic risk scores (PRS)
Individual PRS for GERD, asthma, allergic rhinitis, and eczema were generated in each sub-cohort using summary statistic data from the largest to-date and publicly available GWAS on the relevant diseases (see Table 5 for information on discovery sets).

Searching the GWAS Catalog, Pubmed, MedRxiv, and UK Biobank's website revealed several available summary statistic data for allergic traits and GERD. The analyzed association results were used for each SNP from the discovery sets with the largest sample sizes. First, the largest genome-wide association study of asthma to date (153,763 cases and 1,647,022 controls) was identified via meta-analysis across 18 biobanks spanning multiple countries and ancestries. Specifically, the European ancestry-based summary statistics for asthma from 14 biobanks, i.e., BioMe, BioVU, CCPM, DECODE, ESTBB, FinnGen, GS, HUNT, Lifelines, MGB, MGI, QSKIN, UCLA, UKBB were used as discovery samples to estimate the PRS[40]. However, most of the GWAS on asthma phenotypes are under-powered. For example, Nick Shrine and others have published one study on moderate-to-severe asthma using 5135 cases and 25675 controls from the UK[41]. The eosinophilic asthma phenotype was under-powered in the UK Biobank sample with 2302 cases and 358892 controls. No GWAS reported on allergic asthma. We could only use one available powerful GWAS (based on a UK Biobank sample) on childhood-onset asthma (COA) and adult-onset asthma (AOA) (COA cases: 13,962, AOA cases: 26,582, common set of controls: 300,671)[22].

For allergic rhinitis, the largest genome-wide meta-analysis available and published was chosen (59,762 cases and 152,358 controls) to be the discovery sample[25]. Non-allergic rhinitis with 2028 cases and 9606 controls was not used due to the small sample size. The authors combined data from children 6 years and above as well as adult participants. Allergic rhinitis was defined as individuals having either a diagnosis or symptoms of allergic rhinitis depending on cohort-specific data availability (23andMe, UKBB, deCODE being the largest three studies together with 22 researcher-led cohorts).

Regarding eczema, the summary statistics data from the largest genome-wide meta-analysis on eczema using participants from the Finn-Gen, Estonian Biobank, and the UK Biobank with European ancestry (22,474 cases and 774,187 controls) was chosen[26]. Phenotypic definition for eczema was based on diagnostic records with relevant ICD-codes (ICD-10 L20; ICD-9 6918; ICD-8: 691) among adults.

Regarding GERD, the largest genome-wide meta-analysis of GERD (80,265 cases and 305,011 controls) using population-based samples from the UK, the USA, and Australia was chosen to be the discovery set[21]. Slightly different definitions of GERD phenotype were used in the UK Biobank (diagnosis of GERD by ICD-10 codes, self-reported GERD condition, or use of GERD medication which in total consists of 68,535 cases and 250,910 controls), 23andMe (self-reported doctor diagnosis with heartburn, acid reflux or acid reflux disease, or treated with medicines for acid reflux/heartburn, which in total consists of 8743 GERD cases and 43,932 controls), and QSkin samples (self-reported heartburn or dispensed reflux medications identified from the PBS database, which in total consists of 2987 cases and 10,169 controls).

There was no sample overlap between the twin sub-cohorts and the summary statistic data from discovery sets. We used SBayesR to generate individual PRS for each trait, which has better prediction accuracy compared to other conventional PRS approaches including clumping and thresholding methods[42]. Due to the complex and long-ranging linkage disequilibrium (LD), it is recommended to exclude the major histocompatibility complex (MHC) region when applying SbayesR. However, several of our traits harbor variants of large effects in the MHC region. We therefore first extracted the most significant variant in the MHC region for each trait, applied SbayesR as normal (which excludes the MHC region), and added back in the most significant variant, using the raw effect size from the original GWAS. After obtaining the SBayesR estimates of SNP effects, we used plink2's—score command to generate polygenic risk scores and standardized the scores.

## Statistics and reproducibility
**Bivariate associations between allergic diseases and GERD.** To confirm the phenotypic associations between each allergic disease including asthma with GERD for each twin sub-cohort, we applied generalized estimating equations (GEE) with logit link function and corrected for twin clustering. Models were adjusted for birth year and sex.

**Quantitative genetic analyses.** Individual cross-trait (i.e., phenotypic) correlation, cross-twin within-trait, and cross-twin cross-trait correlations were estimated as tetrachoric correlations for each allergic disease including asthma and GERD. Different univariate and bivariate structural equation models were then tested to quantify the proportion of variation in liability to allergic diseases and GERD that was due to genetic and environmental components[43]. Estimations using classic twin methodology were calculated for (1) additive genetic deviations (noted as A, assuming MZ twins share 100% and DZ twins share 50% of genetic variance); (2a) non-additive/dominant genetic deviations (noted as D, assuming MZ twins share 100% and DZ twins share 25% of the dominance deviation variance); or (2b) shared environmental effects (noted as C, assuming MZ and DZ twins share 100% environmental influences), and (3) unique environmental effects (noted as E)[44]. Univariate ACE (including A, C, and E-sources of variance and covariance), ADE, and AE models were fitted, with adjustments for sex and birth year. For a simplified interpretation of bivariate ADE models, the broad-sense heritability A + D was also calculated. The likelihood ratio test and the AIC were used to select the best-fitting model. Bivariate models were fitted with a 4 × 4 predicted variance-covariance matrix for MZ and DZ pairs to decompose the variation within diseases and covariation between diseases into A, C/D, and E components. Additionally, rA/rC/rD/rE was used to indicate the strength/correlation efficiency of A/C/D/E explaining the phenotypic correlation between diseases. The Wald method was used to calculate 95% CIs for all parameter estimates.

**Polygenic risk score analyses.** First, correlations between the PRS of each trait in each sub-cohort and the relevant phenotypic definitions were estimated to validate the PRS. The PRS prediction accuracy and performance were assessed by using the AUC value, OR by decile (i.e., checking the sign of the logistic regression coefficient in the expected direction), and Nagelkerke Pseudo-$R^2$. Second, GEE with logit link function was used to assess the association between PRS for each allergic disease with

GERD phenotype, and the PRS for GERD with each allergic disease phenotype among all twins with available genotype data. The GEE quasi-likelihood approach modeled the correlated data by specifying an exchangeable working correlation matrix to account for the correlation due to clustering within twin pairs. OR and 95% confidence intervals (CI) were presented without or with adjustment for birth year, sex, and the interaction between the top 5 principal components (population stratifications) with sub-cohort (i.e., STAGE, TwinGene, SALTY). The analyses were performed in SAS 9.4.

**Linkage disequilibrium score regression (LDSC).** The genetic correlation between allergic diseases and GERD was estimated in LDSC using published GWAS summary statistics (see Table 5). Association test statistics were regressed on their LD scores, a measure of each SNP's relationship with other variants. SNP-based heritability ($h^2_{SNP}$ converted to liability scale with population-based prevalence from literature and sample prevalence data from summary statistics) was estimated as well as the genetic correlation ($r_g$) for each allergic disease with GERD using LDSC. Statistical significance was assessed using Bonferroni correction[45]. Analysis was performed using the LDSC command line tool (Python 2.7.5).

**Genomic structural equation model.** Using Genomic Structural Equation Modeling[46] (Genomic SEM) we aimed to assess if a single underlying latent factor could explain the overlap between GERD, asthma, and allergic diseases. Genomic SEM uses GWAS summary statistics to identify factor structures in the genetic correlation pattern between traits. Because of the conceptual and factual genetic overlap (see Supplementary Fig. 1) between asthma and the asthma sub-types (childhood and adult-onset) we fit separate models containing either all-onset asthma or the sub-types. Following common practice, cut-off values for acceptable model fit were set at CFI >0.90 and SRMR <0.10. The loading of the first indicator was fixed at 1. Asthma was a Heywood case in the model containing the all-onset asthma trait, and its variance was forced to be positive (>0.001). The analyses were performed in R version 4.2.3.

**Bidirectional two-sample Mendelian randomization (MR).** Bidirectional two-sample MR analysis was performed to strengthen the causal inference of the results using the TwoSample MR R package (R version 4.2.3)[47]. The SNPs associated with each trait at the genome-wide significance level ($p < 5 \times 10^{-8}$) with clumping window > 10,000 kb and the LD level ($r^2 < 0.001$) were selected as instrumental variables (IV) from published GWAS summary statistics (see Table 5). The instruments' strength in the final IV set was detected with $F$-statistics after the exclusion of palindromic variants. Due to the very high sample overlap with GERD summary statistics, we did not use the asthma subtype summary statistics for MR analyses. We used the multiplicative random effects IVW model as the primary MR method to estimate the associations of genetically determined allergic traits with the risk of GERD, and vice versa. Sensitivity analyses with unweighted and weighted mode-based estimations, weighted median, and MR-Egger methods were performed to examine the robustness of the results and identify horizontal pleiotropy. Leave-one-out analysis was performed to assess whether there was a significant effect on the results after the removal of a single SNP instrument.

**Gene-based association analysis.** To further understand the potential biological mechanisms underlying the genetic associations, published GWAS summary statistics were utilized to run gene-based and/or gene-set-based analysis with MAGMA[48] v1.6 (a dynamic repository for trait-associated gene discoveries) on the FUMA platform[49]. From all the protein-coding genes included in the Ensembl build 85 and tested "Curated gene sets" and "Gene Ontology (GO) terms" included in the Molecular Signatures Database (MSigDB v7.0), the relevant genome-wide significant genes/gene sets for each disease were identified (after Bonferroni correction). Then these results were compared between each allergic disease and GERD to identify possible shared genes/gene-sets for comorbidities. Enrichment of differentially expressed gene (DEG) sets in a specific tissue type compared to the average gene expression by all general tissue types was also presented from MAGMA tissue enrichment analysis.

## Reporting summary
Further information on research design is available in the Nature Portfolio Reporting Summary linked to this article.

## Data availability
The quantitative genetic and polygenic risk score analyses are based on original data held by the Swedish National Board of Health and Welfare, Statistics Sweden, and the Swedish Twin Registry, https://ki.se/en/research/the-swedish-twin-registry. Due to Swedish data storage laws, we cannot make the data publicly available, however, any researcher can access the data by obtaining ethical approval and then asking the registers for the original data. Pseudonymized data may also be provided by the PI upon request if providing a reasonable proposal and if an appropriate data-sharing agreement with Karolinska Institutet can be established. All other analyses use publicly available data.

## Code availability
Codes and scripts used for all statistical analyses can be shared upon request.

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

## Acknowledgements

We acknowledge the Swedish Twin Registry for access to data, Camilla Palm, and Robert Karlsson for the work on quality control and quality assurance in all the individual phenotype and genotype data. The Swedish Twin Registry is managed by Karolinska Institutet and receives funding through the Swedish Research Council under grant no. 2017-00641. We wish to thank the Biobank at Karolinska Institutet for professional biobank service. Financial support was provided by the Swedish Research Council (grant no. 2018-02640 and 2023-02327), the Swedish Heart-Lung Foundation (grant no. 20180512 and 20210416), and Karolinska Institutet (grant no. 2020-0007 and 2022-02303).

## Author contributions

T.G., B.B., and C.A. conceived of the study. T.G., B.B., C.A., R.K.H., C.L., Y.L., N.T., and A.A. designed the study, T.G. and J.P. performed the analyses, A.H. and Y.L. contributed to data preparation, all authors (T.G.,

R.K.H., A.H., C.L., A.S., K.L., A.A., Y.L., N.T., J.P., C.A., B.B.) contributed to the interpretation of results, T.G. and B.B. prepared the first draft of the manuscript, all authors (T.G., R.K.H., A.H., C.L., A.S., K.L., A.A., Y.L., N.T., J.P., C.A., B.B.) approved the final version for submission.

## Funding

## Competing interests
The authors have no conflict of interest to declare except for Professor Nicholas Talley (NJT). NJTs involvement is all outside the submitted work: Norgine (2021)(IBS interest group), personal fees from Allakos (gastroduodenal eosinophilic disease) (2021), twoXAR Viscera Labs, (USA 2021) (IBS-diarrhea), IsoThrive (2021) (esophageal microbiome), BluMaiden (microbiome advisory board) (2021), Rose Pharma (IBS) (2021), Intrinsic Medicine (2022) (human milk oligosaccharide), Comvita Mānuka Honey (2021) (digestive health), Astra Zeneca (2022). In addition, Dr. Talley has a patent Nepean Dyspepsia Index (NDI) 1998, a patent Licensing Questionnaires Talley Bowel Disease Questionnaire licensed to Mayo/Talley, "Diagnostic marker for functional gastrointestinal disorders" Australian Provisional Patent Application 2021901692, "Methods and compositions for treating age-related neurodegenerative disease associated with dysbiosis" US Application No. 63/537,725.
