## [Peer review file · Communications Biology]

Reviewers' comments:

Reviewer #1 (Remarks to the Author):

Thanks the author used Quantitative genetic modeling, polygenic risk scores (PRS), linkage disequilibrium score regression (LDSC) and gene-/gene-set based analysis to explore the phenotypic and genetic correlations between GERD and each of asthma, allergic rhinitis and eczema. Then, the finding of a common genetic architecture for asthma and GERD may be attributable to some of the observed comorbidities of these diseases, but is unlikely to explain eczema, allergic rhinitis and GERD comorbidities.

However, the following issues require further understanding:

1. In the results section, many results are illustrated and presented with supplementary tables or figures. If these results are important, they should be in the main text rather than the supplement.

2. Some data in the article are not clearly explained.

A. "Polygenic associations were observed between asthma and GERD in the range of OR 1.09-1.14 per one SD increase in PRS." in the abstract section, 1.09-1.14 is the result of which table?

B. "Phenotypic associations were confirmed for: asthma and GERD- adjusted odds ratios (adjOR) 1.68, 95% CI 1.43, 1.98; allergic rhinitis and GERD- adjOR 1.19 (95%CI 1.02, 1.39); eczema and GERD- adjOR 1.22 (95%CI 1.01, 1.47)." in the first paragraph of the results section, how are the results calculated and from which table?

3. Some values are different in the article and in the table.

A. In result quantitative genetic analyses section, the article writes "63% for asthma (95%CI 0.58, 0.67), 61% for allergic rhinitis (95%CI 0.57, 0.65)" but the table writes "asthma 95%CI 0.58, 0.68, and allergic rhinitis 95%CI 0.56, 0.65".

A revision is suggested for the presentation and correctness of the Results section.

Reviewer #2 (Remarks to the Author):

The authors provide a cross-trait analysis to measure the genetic component of the shared etiology between GERD and atopic diseases. While long known to be clinically correlated, this represents an important step to understanding the genetics of these complex traits and narrow in on biology.

Major points:

The introduction would benefit from a summary of the literature describing the heritability of these traits (both twin-based estimates and SNP-based estimates). Further discussion of the complex genetics and the complex environmental risk factors would help frame the question of focus. How much of the of the variance in these phenotypes has been explained through the latest PRS approaches?

Further description of the age of onset for these diseases would add important information about how these traits are comorbid. The atopic traits of focus (atopic dermatitis, asthma, allergic rhinitis) have a trends in prevalence by age group – how does GERD fit into these trends?

“GERD in patients with asthma between 17% and 53% depending on the size of study, study population-type and the asthma phenotype”

The range of 17-53% is quite large, and further interpretation and description of the population-type differences and the asthma phenotype difference would provide helpful background. Are there differences by age or other factors?

Applying GenomicSEM or a similar technique to the summary statistics would improve the authors interpretation of how these traits overlap genetically, and has already shown success with correlated complex traits. Does a common factor model fit well for these traits? A exploratory factor analysis and a confirmatory factor analysis would be of interest to test if there are distinct genetic factors.

Applying MiXeR to these summary statistics would also improve the depth of the analysis.

The analysis could benefit from downstream evaluation and discussion of vertical vs horizontal pleiotropy for the genes that are highlighted.

The results shown in Figure S11 should be discussed in the main text

Minor points:

“a number of intervention studies aiming to improve asthma symptoms by using anti-reflux (acid reducing) medication have not been successful”

Have studies tested the effectiveness of using asthma medications to reduce GERD symptoms or in an in vitro/benchtop setting?

A more detailed description of the symptoms of GERD and the atopic traits (and the impact on affected patients quality of life) would help improve the motivation for this study.

A 6x6 heatmap showing the genetic correlations between all traits would be much more informative than what is shown in table 3.

The y-axis label should be fixed in Figure S1

Figure S10 seems to be missing data

Response to *Communications Biology* Reviewers ‘Shared genetic architecture between gastro-esophageal reflux disease, asthma and allergic diseases: application of genetically informative methods’.

Reviewer 1.

1. In the results section, many results are illustrated and presented with supplementary tables or figures. If these results are important, they should be in the main text rather than the supplement.

[Response]

Thank you. We have now moved Table S1 describing the studies from which we used the summary statistics to the main text (new Table 1), and added a new figure, Figure 2, which highlights the new genomic SEM results, as suggested by reviewer #2 (see below). The remaining results are still presented in the supplement due to space limitations and that they mostly focus on subgroup outcomes of secondary importance to the main findings.

2. Some data in the article are not clearly explained.

A. “Polygenic associations were observed between asthma and GERD in the range of OR 1.09-1.14 per one SD increase in PRS.” in the abstract section, 1.09-1.14 is the result of which table?

[Response]

These results come from Figure 1 and Supplementary Table 8. The adjusted odds ratios and 95% confidence intervals have now been added to Figure 1 so that they are in the main manuscript.

B. "Phenotypic associations were confirmed for: asthma and GERD- adjusted odds ratios (adjOR) 1.68, 95% CI 1.43, 1.98; allergic rhinitis and GERD- adjOR 1.19 (95%CI 1.02, 1.39); eczema and GERD- adjOR 1.22 (95%CI 1.01, 1.47)." in the first paragraph of the results section, how are the results calculated and from which table?

[Response]

Thank you. The phenotypic associations were estimated using generalized estimating equation models as described in the statistical analyses section, first paragraph. We realized this was not presented in a separate table as part of this had been published previously (<https://onlinelibrary.wiley.com/doi/full/10.1111/cea.14106>). The bivariate associations between asthma/allergic traits and GERD are now tabulated in Table S2 in the supplementary materials:

Table S2. Bivariate associations between asthma/allergic traits and GERD.

Exposure	Outcome	OR (95% CI)	
		Unadjusted	Adjusted*
Asthma	GERD	1.64 (1.47, 1.84)	1.61 (1.43, 1.80)
No asthma		ref	Ref
Hay fever	GERD	1.15 (1.03, 1.28)	1.14 (1.02, 1.28)
No hay fever		ref	ref

Eczema No eczema	GERD	1.19 (1.05, 1.36) ref	1.14 (1.00, 1.30) ref
GERD No GERD	Asthma	1.61 (1.43, 1.80) ref	1.58 (1.41, 1.77) Ref
GERD No GERD	Hay fever	1.12 (1.01, 1.25) ref	1.13 (1.01, 1.27) ref
GERD No GERD	Eczema	1.19 (1.05, 1.36) ref	1.14 (1.00, 1.31) ref

3. Some values are different in the article and in the table.

In result quantitative genetic analyses section, the article writes “63% for asthma (95%CI 0.58, 0.67), 61% for allergic rhinitis (95%CI 0.57, 0.65)” but the table writes “asthma 95%CI 0.58, 0.68, and allergic rhinitis 95%CI 0.56, 0.65”. A revision is suggested for the presentation and correctness of the Results section.

[Response]

Our reading of the results section paragraph 2 and Table 2 is that they are the same. The results for asthma and allergic rhinitis are read from the AE model in the table. However, in line with the reviewers broader concern, we have checked through the manuscript for all values presented in the main text to align with the tables/figures.

Reviewer #2:

The authors provide a cross-trait analysis to measure the genetic component of the shared etiology between GERD and atopic diseases. While long known to be clinically correlated, this represents an important step to understanding the genetics of these complex traits and narrow in on biology.

Major points:

1. The introduction would benefit from a summary of the literature describing the heritability of these traits (both twin-based estimates and SNP-based estimates). Further discussion of the complex genetics and the complex environmental risk factors would help frame the question of focus. How much of the of the variance in these phenotypes has been explained through the latest PRS approaches?

[Response]

We thank reviewer #2’s suggestion regarding the literature summary on these traits’ heritability. Variance of asthma explained by asthma PRS and eczema explained by eczema PRS in two recent studies have been estimated at 0.02-0.05 and 0.06 (Table S5 of B Namjou et al, JACI 2022; Table E6 of

Arehart CH et al, JACI 2022). Unfortunately, no previous studies have reported the variance explained by allergic-rhinitis and GERD PRS.

We have re-written parts of the introduction to reflect the missing heritability and motivate further the molecular genetic approaches that we used in the study to investigate the shared genetic architecture between GERD with asthma and allergic diseases as suggested by Reviewer 2 (see below). In addition, the Reviewer will note other changes to the introduction as a result of points 2 and 7 below and the descriptions of GERD and atopic diseases.

2. Further description of the age of onset for these diseases would add important information about how these traits are comorbid. The atopic traits of focus (atopic dermatitis, asthma, allergic rhinitis) have a trends in prevalence by age group – how does GERD fit into these trends? “GERD in patients with asthma between 17% and 53% depending on the size of study, study population-type and the asthma phenotype” .The range of 17-53% is quite large, and further interpretation and description of the population-type differences and the asthma phenotype difference would provide helpful background. Are there differences by age or other factors?

[Response]

Thank you for this observation. We have added some extra words to this sentence about the 17-53% comorbidity range in the introduction to highlight why there is such a range in various studies. The sentence now reads:

‘Epidemiological studies report the comorbidity of GERD in patients with asthma between 17% and 53% depending on the country of study (Western countries have higher prevalence of GERD),³ study population-type (patient group/general population, child/adult) and the detection methods used for asthma and GERD (symptoms alone or lab-based monitoring).^{2, 4, 5}

Regarding the issue of age of onset of these diseases. Asthma, eczema, allergic rhinitis and GERD can affect both children and adults, with incident cases at all points in the life course. However, in recognition that atopic diseases are more commonly a childhood disease and GERD more commonly in adulthood we have added these words to the introduction:

‘Atopic diseases are characteristically childhood diseases beginning very early in life, with 20-25% of cases continuing into adulthood as well as new onset of adult cases.²⁰ GERD is predominantly an adult disease, although infants and adolescents can also suffer with GERD symptoms at clinically significant rates.³

3. Applying GenomicSEM or a similar technique to the summary statistics would improve the authors interpretation of how these traits overlap genetically, and has already shown success with correlated complex traits. Does a common factor model fit well for these traits? A exploratory factor analysis and a confirmatory factor analysis would be of interest to test if there are distinct genetic factors.

[Response]

We thank reviewer’s suggestion on Genomic SEM analyses. We have now performed a common factor model which fits well for most traits except for eczema, suggesting that the common latent factor could explain the shared genetic liability to asthma, allergic rhinitis and GERD. This supports our other findings.

The Abstract, Methods, Results and Discussion were accordingly updated in the main text. The additions are shown here including Figure 2 and Supplementary Table S10.

METHODS:

Genomic Structural Equation Model

Using Genomic Structural Equation Modeling³³ (SEM) we aimed to assess if a single underlying latent factor could explain the overlap between GERD, asthma, and allergic diseases. Genomic SEM uses GWAS summary statistics to identify factor structures in the genetic correlation pattern between traits. Because of the conceptual and factual genetic overlap (see Figure S1) between asthma and the asthma sub-types (childhood and adult onset) we only used the general asthma trait in the common factor model. Following common practice, cut-off values for acceptable model fit were set at Comparative Fit Index (CFI) > .90 and Standardized Root Mean Square Residual (SRMR) < .03. The loading of the first indicator (GERD) was fixed at 1. Asthma was a Heywood case in our model, and its variance had to be forced to be positive (>.001).

RESULTS:

Genomic SEM

A single latent factor fit the genetic covariance structure reasonably well, with CFI = .99 and SRMR = 0.04. However, eczema did not load on the common factor, $\beta = -3 \times 10^{-4}$, $p = .995$, suggesting this trait did not cluster together with asthma, GERD, and allergic rhinitis. The standardized loading of asthma and allergic rhinitis exceeded 1 ($\beta = 2.19$, $p = 2 \times 10^{-4}$ and $\beta = 1.11$, $p = 2 \times 10^{-13}$, respectively; Figure 2, Table S10). This could be a result of having few indicators (with one not loading on the factor at all) and should not be a threat to the interpretation of the common factor model.

DISCUSSION

Paragraph 2: ‘. The Genomic SEM results also suggest that the common genetic liability to GERD, asthma and allergic rhinitis can be summarised by a single underlying common factor.’

Paragraph 3: ‘.. neither were eczema-associated SNPs captured in the genetic variance shared between the common factors for GERD and other allergic traits from genomic SEM results.’

Figure 2. Common factor (F1) model for all traits as estimated in Genomic SEM. Standardized path estimates are given with their standard error. The loading of GERD was fixed to 1. Variances are indicated with circular arrows.

* significant, largest p -value $2E-4$

Table S10. Results for the common factor model fit in Genomic SEM.

Trait 1	Operator	Trait 2	β	SE	p -value
Common factor	==	GERD	1	*	*
Common factor	==	Asthma	2.193	0.593	1.99E-4
Common factor	==	Allergic rhinitis	1.115	0.152	2.36E-13
Common factor	==	Eczema	0.000	0.040	0.995
Asthma	~~	Asthma	0.001**	0.261	0.963
GERD	~~	GERD	0.787	0.071	2.07E-28
Allergic rhinitis	~~	Allergic rhinitis	0.735	0.220	0.001
Eczema	~~	Eczema	1.000	0.191	1.54E-07

Common factor	~~	Common factor	0.213	0.061	4.05E-4
---------------	----	---------------	-------	-------	---------

* First indicator was fixed to 1; no SE and p-value are available

** Variance was forced > 0

== factor is measured by; x == y covariance; x == x residual variance

β = standardized genotype; SE = standard error

4. Applying MiXeR to these summary statistics would also improve the depth of the analysis. The analysis could benefit from downstream evaluation and discussion of vertical vs horizontal pleiotropy for the genes that are highlighted.

[Response]

Again, we thank the reviewer's further suggestion on downstream evaluation on pleiotropy.

To provide insight into potential vertical pleiotropy (as well as formally test horizontal pleiotropy) we added Mendelian Randomization analyses to our study. Indeed, we find evidence for a causal effect of GERD on asthma and vice versa, while not observing significant horizontal pleiotropy in the MR Egger analyses. In addition to the MR Egger test, our methods already provide a quite complete picture of horizontal pleiotropy (refer to PRS, genetic correlations and look-up of hits in previous literature) and we deem MiXeR to fall outside of the scope of this work.

Therefore, we decided not to run MiXeR analyses at the moment. The MR analyses have now been added to the methods, results and discussion in the text. We have added them here as well:

METHODS

Bidirectional two-sample Mendelian Randomization (MR)

Bi-directional two-sample MR analysis was performed to strengthen the causal inference of the results using TwoSample MR R package (R version 4.2.3).³⁴ The SNPs associated with each trait at the genome-wide significance level ($P < 5 \times 10^{-8}$) with clumping window > 10,000 kb and the LD level ($r^2 < 0.001$) were selected as instrumental variables (IV) from published GWAS summary statistics (see Table 1). The instruments' strength in the final IV set were detected with F-statistics after exclusion of palindromic variants. Due to the very high sample overlap with GERD summary statistics, we did not use the asthma subtype summary statistics for MR analyses. We used the multiplicative random effects inverse-variance weighted (IVW) model as the primary MR method to estimate the associations of genetically determined each allergic trait with risk of GERD, and vice versa. Sensitivity analyses with unweighted and weighted mode-based estimations, weighted median, and MR-Egger methods were performed to examine the robustness of the results and identify horizontal pleiotropy. Leave-one-out analysis was performed to assess whether there was a significant effect on the results after the removal of a single SNP instrument.

RESULTS

Bidirectional two-sample MR analyses

There was support for a causal effect of genetic liability to asthma on increased risk of GERD, which was consistent across different sensitivity analyses (IVW OR 1.09, 95% CI 1.05-1.14, $p = 6.55 \times 10^{-5}$). In the other direction, similar effect estimates of genetic liability to GERD on increased risk of asthma were also detected (OR 1.27, 95% CI 1.12–1.43, $p = 2.65 \times 10^{-2}$). The MR-Egger regression intercepts did not significantly deviate from zero (Table S11), suggesting no evidence of horizontal pleiotropy. Leave-one-out and Q-heterogeneity analysis showed that the effect estimates were not overly influenced by any one variant (Figures S3 and S6). No support for association between other allergic traits with GERD were observed (Table S11, Figures S4-S8).

DISCUSSION

Paragraph 5: ‘Furthermore, the downstream evaluation by MR results (including one pleiotropic SNP at 12q13) presented with balanced horizontal pleiotropy, meaning the potential causal association between asthma and GERD did not seem to be biased by the two pleiotropic genes mentioned above, but could still be possibly mediated by them through possible inflammatory pathways.’

Table S11. Bidirectional causal relationships between asthma/allergic traits and GERD based on the two-sample Mendelian Randomization analyses. Bonferroni-corrected threshold of $p = 0.008$ (0.05/6 traits).

Exposure-Outcome	Number of SNPs as instrumental variables	Mean F-statistics	MR-Egger intercept	p -value for the MR-Egger intercept test	p -value for MR-Egger Q heterogeneity test	OR (95% CI) by IVW approach	p -value by IVW approach	p -value for IVW Q heterogeneity test
Asthma-GERD	107	74.21	0.005	0.06	0	1.09 (1.05, 1.14)	<0.001	0
Allergic rhinitis-GERD	3	38.92	-0.014	0.58	0.55	0.99 (0.93, 1.05)	0.71	0.62
Eczema-GERD	19	54.44	0.010	0.16	0.03	0.92 (0.70, 1.15)	0.48	0.02
GERD-Asthma	18	33.48	-0.010	0.78	0	1.27 (1.12, 1.43)	0.002	0
GERD-Allergic rhinitis	23	33.96	0.003	0.92	0.17	0.81 (0.58, 1.05)	0.08	0.21
GERD-Eczema	18	33.56	0.006	0.84	0.0003	0.97 (0.75, 1.18)	0.75	0.0005

5. The results shown in Figure S11 should be discussed in the main text

[Response]

Thanks for noticing this. We have now described the results on tissue enrichment analyses and discussed the findings shown in Figures S18-S21 as shown below:

RESULTS:

Last paragraph: 'In the general tissue expression analysis, we found evidence for a small but significant enrichment exclusively in brain tissues for GERD-associated genes; blood, spleen, lung, and small-intestine tissues for asthma; and spleen, blood, and small intestine tissues for eczema, suggesting different tissues are responsible for GERD-gene and upregulated asthma-/eczema- gene signals respectively. (Figures S18-S21)'

DISCUSSION:

Paragraph 6 : 'Using the MAGMA tissue enrichment analysis to detect differentially expressed gene sets for each disease we were expecting to find an overlapping tissue type for GERD and asthma or allergic rhinitis which may provide indication of a candidate gene set to explain genetic overlap. However, the results pointed to different tissue types, asthma gene sets were expressed in blood, spleen, small intestine, and as expected, in lung tissues. GERD genes were most expressed in brain tissues. Rather than shared expression in a shared tissue type it may be that shared genes for asthma and GERD behave differently in each tissue for each disease, ie that genes in blood and spleen lead to inflammatory processes causing asthma, and that in the brain they play a sensory role in reflux pain.'

Minor points:

**6. "a number of intervention studies aiming to improve asthma symptoms by using anti-reflux (acid reducing) medication have not been successful"
Have studies tested the effectiveness of using asthma medications to reduce GERD symptoms or in an in vitro/benchtop setting?**

[Response]

As far as we are aware and after a search there are no studies in vivo or in vitro that have looked at asthma medications to reduce GERD symptoms, in fact, the only reports suggest that some asthma medications may actually increase GERD symptoms, although as these studies are older it seems likely that this phenomenon is not reported with newer medications.

7. A more detailed description of the symptoms of GERD and the atopic traits (and the impact on affected patients quality of life) would help improve the motivation for this study.

[Response]

Thanks for this comment. We have added more words about the symptoms of GERD and each of the atopic traits to the beginning of the introduction:

'Asthma is a common inflammatory respiratory disease causing acute dyspnoea and wheezing, affecting 4-9% of the global child and adult population.¹ The most common non-allergic comorbidity of asthma is gastro-esophageal reflux disease (GERD), characterized by the reflux of

gastric acid into the esophagus causing symptoms such as heartburn and regurgitation often leading to esophagitis and complications such as Barrett's esophagus.^{2'}

8. A 6x6 heatmap showing the genetic correlations between all traits would be much more informative than what is shown in table 3.

[Response]

Thanks. We have now included a 6 x6 heatmap in Figure S1 to illustrate the genetic correlation on the observational scale. However, we believe the information in Table 3 are still important as they include the sample v population prevalence, and the genetic correlations on the liability scale considering some imbalance on the case control sample inclusion of the published GWAS.

9. The y-axis label should be fixed in Figure S1

[Response]

We have fixed the y-axis labels and updated the Figure S1 (now Figure S2).

10. Figure S10 seems to be missing data

[Response]

Thank you for noticing the missing data. Indeed, FUMA could no longer provide the regional plots. We have now replaced all the regional plots using LocusZoom.

Please see the updated Figures S13-17 in the supplementary materials.

Reviewers' comments:

Reviewer #1 (Remarks to the Author):

Thank you for revising, editing, and rewriting. The paper is in pretty good shape.

Reviewer #3 (Remarks to the Author):

The authors have greatly improved various parts of their study with the additions of key descriptions throughout the text, a Genomic SEM common-factor model, and a bidirectional two-sample MR analysis. Despite these improvements, I have a few major concerns that should be addressed before publication.

Major Points:

1. I am pleased to see that the authors have included Genomic SEM common-factor GWAS as a method into their analysis, as it strengthens the manuscript. However, I do have a few concerns regarding the model as it is currently presented. While the CFI = .99 and SRMR = 0.04 suggest good model fit, it is peculiar that the authors have found no genetic correlations for eczema with the other related traits (which are known to be genetically correlated), and it is puzzling that the loading of eczema onto the common factor is zero. It appears the estimated genetic correlations shown in the bottom row of Supplementary Figure 1 are much too low, as they are all between -0.04 and 0.02. The atopic GWAS literature has shown that eczema has a significant non-zero genetic correlation with asthma and allergic rhinitis (e.g., Table 1 from PMID: 32373153 shows genetic correlations of 0.45 for asthma and eczema, and 0.33 for hay fever and eczema; the genetic overlap between these traits has been shown in multiple other publications such as PMID: 31361310 or 33436162). I am worried that the eczema summary statistics were not munged/incorporated into the Genomic SEM analysis properly. Given the results shown in Table 4, it seems as though all of the indicators should have positive loadings onto the common factor for the included traits. As the authors mentioned in the manuscript, it is slightly concerning that the standardized loading for asthma on the common factor is 2.19, as standardized loadings are usually less than or equal to 1. I wonder if correcting eczema in the model will fix the standardized loadings? It is difficult to further comment on the Genomic SEM portion of the manuscript until this potential bug in the analysis has been addressed.

2. The authors have significantly improved the written presentation in many parts of the manuscript. The introduction reads much more clearly and provides helpful

background and context for their study. In addition, the authors now present new results from the new analyses they performed. The discussion, however, is insufficient in its current form and a similar round of revisions should be performed to clarify the conclusions and take-aways from the results. As the authors point out, a primary strength of their paper stems from the triangulation of evidence from different approaches. However, as a reader I am lacking an explanation of the meaning and value of the research findings. Below I list out a few areas of the discussion that could be developed more fully:

a. In Figure 1, the reported difference in predictive utility between allergic rhinitis PRS and GERD phenotype, vs GERD PRS and allergic rhinitis phenotype is interesting but isn't discussed. Synthesizing the predictive results from Figure 1 with the results from the bi-direction MR analysis would be helpful.

b. The results from Table 2 (the twin models) could be discussed at greater depth. There are differing twin-model estimates across the 4 phenotypes, despite modest differences in SNP-based heritability. How does the twin model help us make more sense of these complex phenotypes? How do the results connect to the literature review in the introduction regarding disease onset and comorbidity?

c. The authors suggest that the genetic enrichment of genes in the brain might be "linked to a sensory role in reflux pain." The authors may want to consider digging deeper into this finding and consider additional potential links, such as:

i. There are comorbidities of internalizing psychopathology disorders, asthma, and GERD (note that asthma and anxiety have a genetic correlation of 0.406, asthma and MDD have a genetic correlation of 0.215, and asthma and PTSD have a genetic correlation of 0.458 in Table 1 of PMID: 31619474)

ii. The brain has an important role in behavioral traits, such as alcohol use and smoking which both increase the risk of GERD.

iii. The brain has a key role in diet which is an important factor for GERD, and there is a genetic link to how "food-liking traits correlate with different brain areas and other food consumption traits" (PMID: 35585065).

d. The three examples included above are not an exhaustive list, but they hopefully illustrate a few ways in which the discussion could be strengthened. The discussion needs revisions that emphasize the ways in which the authors' triangulation of methods and results reveals novel insights into the understanding of GERD in the context of allergic diseases.

3. For the PRS analysis, it would be helpful to understand the incremental increase in prediction accuracy (“When covariates are included in the model, then measures such as the incremental R² [increase in R² with the addition of the PRS to the model], which isolate the explanatory power of the PRS, should be reported.” PMID: 32709988).

4. For the PRS analysis, I am concerned about the relatedness in the testing dataset (given that the samples are from a twin study). In the supplement, the authors briefly summarize that they used a “series [of] logistic regression models with clustered standard errors.” I suggest the authors expand on how they handled sample relatedness when constructing these PRSs (e.g. LD structure) and when evaluating these PRSs (assessing AUC and OR).

Minor Points

1. Figure S2 should be reformatted since it has overlapping y-axis labels and cropped titles.

2. The Genomic SEM common factor could be used as a training dataset for an additional PRS to be shown in Figure 1 (as was done in Figure 3 of PMID: 30962613). This might help tie together the authors’ analyses into a more unified framework.

Response to the reviewers 'Shared genetic architecture between gastro-esophageal reflux disease, asthma and allergic diseases: application of genetically informative methods'.

Reviewer #1 (Remarks to the Author):

Thank you for revising, editing, and rewriting. The paper is in pretty good shape.

Reviewer #3 (Remarks to the Author):

The authors have greatly improved various parts of their study with the additions of key descriptions throughout the text, a Genomic SEM common-factor model, and a bidirectional two-sample MR analysis. Despite these improvements, I have a few major concerns that should be addressed before publication.

[Response]

Thank you for these comments. Please see our responses below.

Major Points:

1. I am pleased to see that the authors have included Genomic SEM common-factor GWAS as a method into their analysis, as it strengthens the manuscript. However, I do have a few concerns regarding the model as it is currently presented. While the CFI = .99 and SRMR = 0.04 suggest good model fit, it is peculiar that the authors have found no genetic correlations for eczema with the other related traits (which are known to be genetically correlated), and it is puzzling that the loading of eczema onto the common factor is zero. It appears the estimated genetic correlations shown in the bottom row of Supplementary Figure 1 are much too low, as they are all between -0.04 and 0.02. The atopic GWAS literature has shown that eczema has a significant non-zero genetic correlation with asthma and allergic rhinitis (e.g., Table 1 from PMID: 32373153 shows genetic correlations of 0.45 for asthma and eczema, and 0.33 for hay fever and eczema; the genetic overlap between these traits has been shown in multiple other publications such as PMID: 31361310 or 33436162). I am worried that the eczema summary statistics were not munged/incorporated into the Genomic SEM analysis properly. Given the results shown in Table 4, it seems as though all of the indicators should have positive loadings onto the common factor for the included traits. As the authors mentioned in the manuscript, it is slightly concerning that the standardized loading for asthma on the common factor is 2.19, as standardized loadings are usually less than or equal to 1. I wonder if correcting eczema in the model will fix the standardized loadings? It is difficult to further comment on the Genomic SEM portion of the manuscript until this potential bug in the analysis has been addressed.

[Response]

Thank you for the concern about the eczema GWAS sumstat munging. We have now checked the eczema GWAS summary data downloading and munging processes. The harmonized data from the GWAS Catalog (downloaded via http://ftp.ebi.ac.uk/pub/databases/gwas/summary_statistics/GCST90027001-GCST90028000/GCST90027161/harmonised/34454985-GCST90027161-EFO_0000274.h.tsv.gz) was

originally used in our LDSC and Genomic SEM analysis, and indicated no genetic correlation between asthma and eczema or no loading. However, we observed a moderate genetic correlation when utilizing the non-harmonized summary data (downloaded via http://ftp.ebi.ac.uk/pub/databases/gwas/summary_statistics/GCST90027001-GCST90028000/GCST90027161/GCST90027161_buildGRCh38.tsv.gz, please see Table R1 below).

Therefore, we have chosen to use the author-uploaded non-harmonized summary data, re-run the LDSC and genomic SEM analyses and updated the results accordingly.

We have also removed some of the sentences in the Results under Genomic SEM and changed to:

“All traits loaded significantly on the common factor, with the strongest loading for eczema ($\beta=0.85$, $SE=0.10$, $p=1E-33$) and the lowest loading for GERD ($\beta=0.24$, $SE=0.03$, $p=2E-14$; Figure 2, Table S10). Using the combined asthma trait instead of childhood onset and adult onset separately deteriorated fit, with SRMR falling short of the <0.10 criterion (CFI=0.96, SRMR=0.15).”

Table R1. SNP-based heritability (h^2_{SNP}) and genetic correlation (r_g) estimates of asthma and eczema.

	N (sample prevalence %, population prev. %)	Nr. of SNPs remained in the analysis	h^2_{SNP} (CI) ¹	Intercept (SE) ²	Ratio (SE) ³
Univariate analyses					
Asthma	1800785 (8.5%, 8%)	958562	0.079 (0.071, 0.087)	1.0886 (0.0122)	0.1162 (0.016)
Eczema – harmonized data	796661 (2.8%, 3%)	1168494	0.033 (0.021, 0.045)	1.0423 (0.0088)	0.3549 (0.0736)
Eczema – not harmonized data	796661 (2.8%, 3%)	1168494	0.033 (0.020, 0.046)	1.0423 (0.0088)	0.3549 (0.0736)
Bivariate analyses					
	h^2_{SNP} (trait 1): h^2_{SNP} (trait 2)	r_g (CI)	p-value	Intercept (trait 1): Intercept (trait 2)	Ratio (trait 1): Ratio (trait 2)
Asthma-Eczema -harmonized	0.0626: 0.1329	0.0003 (-0.0401, 0.0409)	0.9883	1.0659:1.0401	0.086: 0.345
Asthma-Eczema -not harmonized	0.0827:0.0326	0.633 (0.524, 0.742)	3.226e-30	1.0658:1.0402	0.0861: 0.3464

2. The authors have significantly improved the written presentation in many parts of the manuscript. The introduction reads much more clearly and provides helpful background and context for their study. In addition, the authors now present new results from the new analyses they performed. The discussion, however, is insufficient in its current form and a similar round of revisions should be performed to clarify the conclusions and take-aways from the results. As the authors point out, a primary strength of their paper stems from the triangulation of evidence from different approaches. However, as a reader I am lacking an explanation of the meaning and value of the research findings. Below I list out a few areas of the discussion that could be developed more fully:

a. In Figure 1, the reported difference in predictive utility between allergic rhinitis PRS and GERD phenotype, vs GERD PRS and allergic rhinitis phenotype is interesting but isn't discussed. Synthesizing the predictive results from Figure 1 with the results from the bi-direction MR analysis would be helpful.

[Response]

Thank you for the opportunity to review our manuscript and provide suggestions to improve the explanation of our research findings in the discussion. As you state, we observed GERD-PRS associated with allergic rhinitis but no association between allergic rhinitis-PRS and GERD. Furthermore, there is little evidence for genetic correlation based on LDSC or the causal association between GERD and allergic rhinitis based on the bi-directional MR analyses. Therefore, we think that the association of GERD-PRS and allergic rhinitis could be possibly false positive due to the number of analyses and the lack of consistency in the direction of effects/correlation in PRS, LDSC, and MR-analyses.

Therefore, we have revised our discussion which now reads “Allergic rhinitis and GERD were weakly associated (phenotypic correlation=0.04), genetic association between the two was supported by the PRS analysis for GERD-PRS and allergic rhinitis phenotype. However, the lack of consistency in the direction of effects measured by the cross-twin cross trait correlations, LDSC regression and bidirectional MR analysis, suggests no evidence of a causal relationship between allergic rhinitis and GERD”.

b. The results from Table 2 (the twin models) could be discussed at greater depth. There are differing twin-model estimates across the 4 phenotypes, despite modest differences in SNP-based heritability. How does the twin model help us make more sense of these complex phenotypes? How do the results connect to the literature review in the introduction regarding disease onset and comorbidity?

[Response]

We agree that “missing heritability” of the phenotypes based on our univariate twin models and SNP-based heritability estimates is important. However, our study focus was not to investigate the components or explanations of the missing heritability. Therefore, we have added the issue of missing heritability of complex traits to the limitations of the discussion section, which now reads:

“Finally, there may be misclassification of phenotypes used in current GWAS due to the need to restrict time windows to boost sample size, thus diluting accuracy. As a consequence, in the context of heterogeneous and highly prevalent diseases like allergic rhinitis, eczema, and GERD, GWAS-identified

common variants only explained a small part of the heritability compared to the moderate-to-high heritability found in twin studies. ”

c. The authors suggest that the genetic enrichment of genes in the brain might be “linked to a sensory role in reflux pain.” The authors may want to consider digging deeper into this finding and consider additional potential links, such as:

i. There are comorbidities of internalizing psychopathology disorders, asthma, and GERD (note that asthma and anxiety have a genetic correlation of 0.406, asthma and MDD have a genetic correlation of 0.215, and asthma and PTSD have a genetic correlation of 0.458 in Table 1 of PMID: 31619474)

ii. The brain has an important role in behavioral traits, such as alcohol use and smoking which both increase the risk of GERD.

iii. The brain has a key role in diet which is an important factor for GERD, and there is a genetic link to how “food-liking traits correlate with different brain areas and other food consumption traits” (PMID: 35585065).

[Response]

Thank you for great suggestions on the potential interpretation based on the gene enrichment analysis. We have incorporated these references and potential interpretations in the discussion which now reads

“The brain is known to play an important role in behavioural traits such as smoking and food consumption; choices which are important factors for GERD development (1). Alternatively, expressed genes may be working through other pathways such as internalizing psychopathology disorders which are known comorbidities for both asthma (2, 3) and GERD (4). Indeed, earlier work by this team and others looking at asthma and GERD comorbidity found that affective traits (depression, anxiety and neuroticism) were important confounders of GERD and allergic disease associations (5, 6).”

d. The three examples included above are not an exhaustive list, but they hopefully illustrate a few ways in which the discussion could be strengthened. The discussion needs revisions that emphasize the ways in which the authors’ triangulation of methods and results reveals novel insights into the understanding of GERD in the context of allergic diseases.

[Response]

We have now updated the interpretation on asthma-GERD results as suggested above and revised the paragraph on GERD-and other allergic diseases, improving the triangulation of methods.

The discussion now reads “Allergic rhinitis and GERD were weakly associated (phenotypic correlation=0.04), and a genetic association between GERD-PRS and allergic rhinitis phenotype was supported by the PRS analysis. However, the lack of consistency in the direction of effects measured by the cross-twin cross trait correlations, LDSC regression and bidirectional MR analysis, suggests no evidence of a causal relationship between allergic rhinitis and GERD. Similarly, eczema and GERD

estimates were null for bidirectional MR analyses, not supporting a causal connection either. Meanwhile, the LDSC regression revealed a possible signal for genetic overlap between eczema and GERD using summary data, which we could not replicate with individual-level data."

And at the end of the same paragraph:

"Therefore, although the triangulation of genetic methods in our study does not support a genetic explanation for GERD-eczema and GERD-allergic rhinitis comorbidity, future research harnessing larger, more accurately defined cohorts will provide further clarification."

3. For the PRS analysis, it would be helpful to understand the incremental increase in prediction accuracy ("When covariates are included in the model, then measures such as the incremental R² [increase in R² with the addition of the PRS to the model], which isolate the explanatory power of the PRS, should be reported." PMID: 32709988).

[Response]

We understand your suggestion and we did follow Choi et al's protocol (7) where it was possible (i.e. for the purpose of reporting Nagelkerke R² and AUC for each trait as a way to "validate" the predicting possibility in each target sample using logistic regression with clustered standard error for twin pairs, described in supplementary methods and shown in supplementary Table S7).

Additionally, there is no true R² for logistic regression models (8) or generalized estimating equation (GEE). Therefore, we have run the PRS analysis using logistic regression with clustered standard errors which takes into consideration the non-independence of twins and reported the pseudo R² and incremental pseudo R² (see Table R2 below). As expected, the risk estimates based on logistic regression analyses were almost identical to those based on GEE (see Table S8).

Therefore, considering the difficulty to interpret the pseudo R² statistics in non-linear models (9), we prefer reporting our main PRS analysis estimating the population average effect of PRS-trait 1 on phenotypic trait 2 based on GEE models with cluster robust standard errors. However, we are open for suggestions if the editors prefer that we also present the Table R2 below.

Table R2. Association between GERD-PRS with allergic diseases and between allergic disease-PRS with GERD among genotyped twins using logistic regression models with clustered standard errors.

Phenotype	PRS	OR (95% CI)				
		Model 1*	Pseudo R ²	Model 2#	Pseudo R ²	Incremental pseudo R ²
Asthma	GERD	1.14 (1.09, 1.20)	0.0023	1.14 (1.08, 1.20)	0.0143	0.0120
Allergic rhinitis	GERD	1.08 (1.03, 1.13)	0.0008	1.08 (1.03, 1.13)	0.0375	0.0367
Eczema	GERD	1.01 (0.96, 1.07)	0.0000	1.01 (0.96, 1.07)	0.0037	0.0037
GERD	Asthma	1.09 (1.05, 1.14)	0.0011	1.10 (1.06, 1.14)	0.0043	0.0032
GERD	COA	0.99 (0.95, 1.03)	0.0000	0.99 (0.95, 1.03)	0.0031	0.0031
GERD	AOA	1.04 (1.00, 1.08)	0.0002	1.04 (1.00, 1.09)	0.0033	0.0031
GERD	Allergic rhinitis	0.99 (0.95, 1.03)	0.0000	0.99 (0.95, 1.03)	0.0031	0.0031
GERD	Eczema	1.02 (0.98, 1.06)	0.0000	1.02 (0.98, 1.06)	0.0031	0.0031

*Model 1 with no adjustment.

#Model 2 adjusted for birth year, sex, cohort, top 5 principal components, and cohort*top 5 principal components.

4. For the PRS analysis, I am concerned about the relatedness in the testing dataset (given that the samples are from a twin study). In the supplement, the authors briefly summarize that they used a “series [of] logistic regression models with clustered standard errors.” I suggest the authors expand on how they handled sample relatedness when constructing these PRSs (e.g. LD structure) and when evaluating these PRSs (assessing AUC and OR).

[Response]

Thank you for this comment. We used SBaysR to optimize the SNP-based effect sizes from the GWAS sumstats, using the pre-computed Sparse LD matrices provided by the GCTB website (see <https://cnsgenomics.com/software/gctb/#LDmatrices>). Plink was used to calculate the raw scores for each individual. Once an individual's PRS was calculated, we assigned the same PRS for the MZ co-twins who were not directly genotyped. The reviewer is correct that PRS estimates (as well as the outcomes of interest) are indeed correlated for a twin and his/her co-twin. Therefore, when running the main cross-trait association analysis between PRS and phenotype, we used GEE to estimate parameters from clustered data due to twinship, i.e. the estimates of variance of the estimated coefficient were adjusted for the clustering of twins within a pair. The standard errors were estimated using robust sandwich estimators. This method has been widely used to estimate the population averaged effect for clustered data (10).

As a result, we have further clarified the methods in the manuscript, which now reads “Second, generalized estimating equations (GEE) with logit link function were used to assess the association between PRS for each allergic disease with GERD phenotype, and the PRS for GERD with each allergic disease phenotype among all twins with available genotype data. The GEE quasi-likelihood approach modelled the correlated data by specifying an exchangeable working correlation matrix to account for the correlation due to clustering within twin pairs. Sandwich estimators correcting for the clustering within twin pairs were applied to standard errors”.

Minor Points

1. Figure S2 should be reformatted since it has overlapping y-axis labels and cropped titles.

[Response]

Thank you. We have revised this figure.

2. The Genomic SEM common factor could be used as a training dataset for an additional PRS to be shown in Figure 1 (as was done in Figure 3 of PMID: 30962613). This might help tie together the authors' analyses into a more unified framework.

[Response]

We agree that a PRS based on a GWAS on the common factor we identified using Genomic SEM would nicely connect the different parts of the paper. Given that the common factor was not part of the original plan or the focus of this paper though, we have decided to not include these extensive additional analyses. The expected benefit of running a common factor GWAS and deriving the PRS would be limited,

because of foreseeable problems associated with extracting results from Genomic SEM and feeding them into another algorithm. Information on issues such as (control-limited) sample overlap between the traits or Heywood cases in the model will not be considered when looking only at the SNP effects on the factor, so that the resulting GWAS can show unpredictable statistical artifacts, such as inflated heritability estimates or unexpected genetic correlations with other traits. Though doable, it would be beyond the scope of the current work to carefully tackle such issues to derive a reliable PRS.

Reference

1. Zhang M, Hou Z-K, Huang Z-B, Chen X-L, Liu F-B. Dietary and Lifestyle Factors Related to Gastroesophageal Reflux Disease: A Systematic Review. *Therapeutics and Clinical Risk Management*. 2021;17(null):305-23.
2. Lehto K, Pedersen NL, Almqvist C, Lu Y, Brew BK. Asthma and affective traits in adults: a genetically informative study. *The European respiratory journal*. 2019;53(5).
3. Zhu Z, Zhu X, Liu CL, Shi H, Shen S, Yang Y, et al. Shared genetics of asthma and mental health disorders: a large-scale genome-wide cross-trait analysis. *Eur Respir J*. 2019;54(6).
4. Zamani M, Alizadeh-Tabari S, Chan WW, Talley NJ. Association Between Anxiety/Depression and Gastroesophageal Reflux: A Systematic Review and Meta-Analysis. *Am J Gastroenterol*. 2023;10.14309/ajg.0000000000002411.
5. Brew BK, Almqvist C, Lundholm C, Andreasson A, Lehto K, Talley NJ, et al. Comorbidity of atopic diseases and gastro-oesophageal reflux: evidence of a shared cause. *Clin Exp Allergy*. 2022;52(7):868-77.
6. Zamani M, Alizadeh-Tabari S, Chan WW, Talley NJ. Association Between Anxiety/Depression and Gastroesophageal Reflux: A Systematic Review and Meta-Analysis. *Am J Gastroenterol*. 2023;118(12):2133-43.
7. Choi SW, Mak TS, O'Reilly PF. Tutorial: a guide to performing polygenic risk score analyses. *Nat Protoc*. 2020;15(9):2759-72.
8. Hu B, Shao J, Palta M. Pseudo-R2 in logistic regression model. *Statistica Sinica*. 2006;16(3):847-60.
9. Sapra RL. Using R2 with caution. *Current Medicine Research and Practice*. 2014;4(3):130-4.
10. Aloisio KM, Swanson SA, Micali N, Field A, Horton NJ. Analysis of partially observed clustered data using generalized estimating equations and multiple imputation. *Stata J*. 2014;14(4):863-83.

REVIEWERS' COMMENTS:

Reviewer #3 (Remarks to the Author):

I appreciate the careful and thoughtful revisions the authors made to the paper. The paper is notably clearer. The revisions address my concerns and feedback and I think the manuscript is ready to accept for publication.